# Unifying Brain Age Prediction and Age-Conditioned Template Generation with a Deterministic Autoencoder

Pauline Mouches[1,2]                                        pauline.mouches@ucalgary.ca
Matthias Wilms[1,3]                                         matthias.wilms@ucalgary.ca
Deepthi Rajashekar[1,2]                                     deepthi.rajasheka1@ucalgary.ca
Sönke Langner[4]                                            soenke.Langner@med.uni-rostock.de
Nils D. Forkert[1,3]                                        nils.forkert@ucalgary.ca

[1] *Department of Radiology and Hotchkiss Brain Institute, University of Calgary, Canada.*

[2] *Biomedical Engineering Program, University of Calgary, Canada.*

[3] *Alberta Children's Hospital Research Institute, University of Calgary, Canada.*

[4] *Institute for Diagnostic Radiology and Neuroradiology, Rostock University Medical Center, Germany.*

## Abstract

Age-related morphological brain changes are known to be different in healthy and disease-affected aging. Biological brain age estimation from MRI scans is a common way to quantify this effect whereas differences between biological and chronological age indicate degenerative processes. The ability to visualize and analyze the morphological age-related changes in the image space directly is essential to improve the understanding of brain aging. In this work, we propose a novel deep learning based approach to unify biological brain age estimation and age-conditioned template creation in a single, consistent model. We achieve this by developing a deterministic autoencoder that successfully disentangles age-related morphological changes and subject-specific variations. This allows its use as a brain age regressor as well as a generative brain aging model. The proposed approach demonstrates accurate biological brain age prediction, and realistic generation of age-conditioned brain templates and simulated age-specific brain images when applied to a database of more than 2000 subjects.

**Keywords:** Brain aging, Deep learning, Generative models

## 1. Introduction

Healthy brain aging in adults is known to be associated with atrophy, whereas different cerebral regions are affected non-uniformly by aging (Peters, 2006). Many neurological diseases, such as Alzheimer's disease (Fotenos et al., 2005) or epilepsy (Whelan et al., 2018), are known to cause increased or accelerated regional or global atrophy. Thus, understanding healthy brain aging is important to improve the early diagnosis of neurological diseases.

For these reasons, multiple studies used the biological brain age as a biomarker for neurological diseases by comparing it to the true chronological age of the subject (Gaser et al., 2013). Biological brain age is usually estimated using structural magnetic resonance imaging (MRI) scans (Cole et al., 2017) or extracted features (Valizadeh et al., 2017) utilizing classical, unidirectional regression models that map an input to an age scalar. However, such models usually lack of interpretability as they are unidirectional. Post-hoc explainability methods, such as gradient-based methods, can be used along with convolutional neural

networks (CNN) to identify the most important regions within the input image (Levakov et al., 2020). However, these methods often lead to noisy outputs and lack of robustness (Arun et al., 2020), introducing the need for interpretable models. On the other hand, recent studies introduced counterfactuals, consisting in generating hypothetical alternates of the reality. Pawlowski et al. (2020), for instance, used a structural causal model to generate brain MRI scan counterfactuals based on age, sex and brain ventricle volume. Their model allows to modify these variables and generate simulated brain MRI scan for specific subjects at a different age, for example, but is not a predictive model. Furthermore, Bass et al. (2020) used image-to-image translation to translate brain MRI scans between two classes corresponding to young and old subjects, while generating class-relevant feature attribution maps.

More recently, bidirectional approaches have been proposed, which allow to estimate biological brain age based on MRI scans and to generate simulated brain MRI scans for specific ages with the same model (Zhao et al., 2019; Wilms et al., 2020). The assumed benefits of such unified models are an improved performance of both tasks, explainability of the unidirectional results as they can be tracked back to the image space, and better use of the training data (Wilms et al., 2020). These bidirectional models map the input MRI scans to a structured latent space, in which one component contains the age-related information while the others contain the remaining subject-specific information. However, only a few studies have built such unified models so far and they used complex stochastic generative models.

In Wilms et al. (2020) for example, an invertible neural network (INN) based framework is used to predict age and reconstruct full MRI scans. The framework results in reasonable age prediction accuracy value and realistic age-conditioned image synthesis. However, the proposed approach requires multiple advanced preprocessing steps and uses deformation fields instead of the MRI scans as inputs to the INN. Moreover, INNs are stochastic generative models with a complex training process that requires a large amount of data and careful parameter tuning.

The main hypothesis of this work is that a deterministic autoencoder can be used to built such a bidirectional, unified brain aging model, which avoids many of the problems related to training and designing stochastic models, and can use minimally pre-processed MRI scans as input. This idea is supported by recent work from the machine learning community (Ghosh et al., 2020), where it was shown that traditional deterministic autoencoders can successfully serve as generative models for many tasks while producing comparable results to, for example, complex stochastic variational autoencoders.

Our model uses a deterministic autoencoder coupled with an invertible latent space disentanglement module to transform the input MRI datasets into a low dimensional latent space, in which the age component is disentangled from the age-unrelated anatomical variations. This disentanglement is achieved with a subspace projection method based on (Li et al., 2020) that takes as its input the latent space of the autoencoder. We show that using such an easily trainable architecture leads to accurate age predictions and allows to generate realistic age-specific templates of the brain. Additionally, age-conditioned generative modeling is achieved by imposing a Gaussian mixture model on the age-unrelated anatomical components of the latent space and fixing the age component.

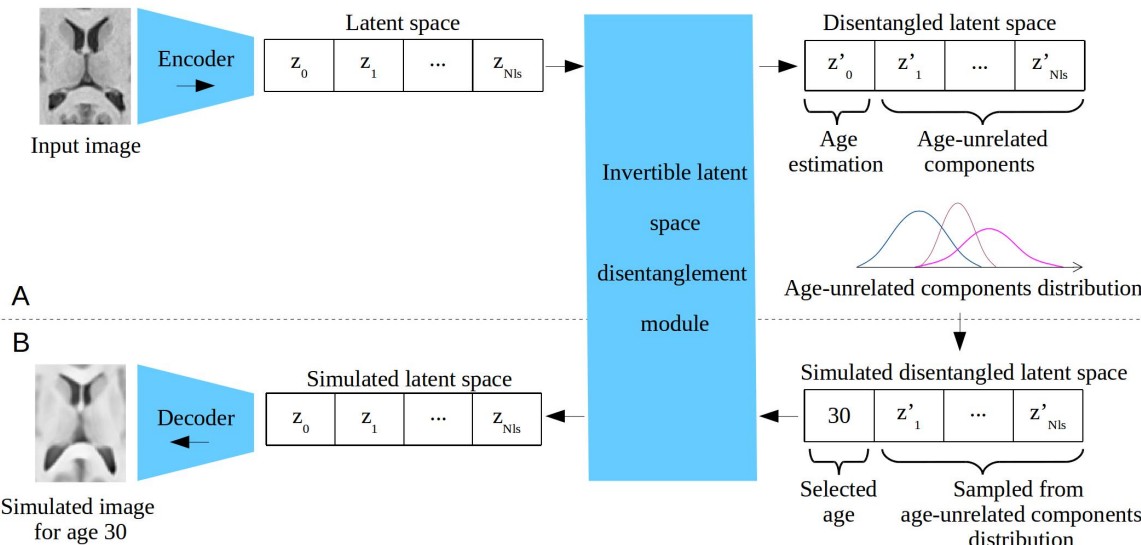

Figure 1: Graphical diagram of the proposed model. A) Latent space disentanglement and biological brain age estimation: The input MRI datasets are encoded into a latent space $z$ which is then transformed into $z'$ where the age-related component is disentangled from the age-unrelated components. The age-unrelated components distribution is estimated using the training datasets. B) Age-specific image simulation: Simulated disentangled latent space representations can be sampled and used to reconstruct age-specific simulated MRI datasets.

## 2. Methods

### 2.1. Problem formulation

In this work, the aim is to train a unified deep learning model able to perform brain age prediction and to generate new samples conditioned on age (see Figure 1). The data used in this study contains for each participant $i \in \{1, ..., N_p\}$:

- a T1-weighted 3D brain MRI scan, cropped around the ventricular region, of size $h \times w \times d$, denoted $X_i \in \mathbb{R}^{h \times w \times d}$.

- the participant's chronological age $a_i \in \mathbb{R}$.

The main component of our model is a deterministic autoencoder consisting of an encoder $e$ that maps an input image to a low dimensional latent space and a decoder $d$, which approximates the inverse operation. Both, $e$ and $d$, are non-linear functions. That is, $e(X_i) = z_i \in \mathbb{R}^{N_{ls}}$, where $N_{ls}$ is the latent space size, and $d(z_i) = \hat{X}_i$, where $\hat{X}_i$ corresponds to the model's reconstruction of $X_i$. In this standard setup, the latent space representation $z_i$ contains age-related and unrelated information about $X_i$. To isolate the age information, we introduce an invertible linear function $f(z_i) = z'_i \in \mathbb{R}^{N_{ls}}$ that maps $z_i$ to a structured representation $z'_i$. Here, the first component of $z'_i$ represents the estimated biological brain age

$\hat{a}_i$, solving the age regression problem, and the other components contain the age-unrelated anatomical variability. With a slight abuse of notation and ignoring the components of $z_i'$ containing the variations unrelated to age, we can write $\hat{a}_i = f \circ e(X_i)$ and $\hat{X}_i = d \circ f^{-1}(a_i)$. We argue that $f$ can be a simple linear function because $e$ and $d$ fully account for the non-linear component needed to perform the mapping (Li et al., 2020). Finally and following (Ghosh et al., 2020), a Gaussian mixture model is fitted to the training data to estimate the distribution of the components of $z_i'$ that are unrelated to age. This allows us to sample new age-conditioned structured representations $z_i'$ from the model and to reconstruct the associated MRI scan.

### 2.2. Model optimization

Model optimization is done by minimizing two losses using the mean squared error (MSE):

1. A standard autoencoder loss penalizing the difference between the original and reconstructed images:
$$L_{ae} = MSE(X_i, \hat{X}_i)$$

2. The latent space disentanglement module loss (Li et al., 2020) penalizing the difference between the predicted biological age and the true chronological age, and the difference between the original and reconstructed latent spaces:
$$L_{ls} = MSE(a_i, \hat{a}_i) + MSE(z_i, f^{-1} \circ f(z_i))$$

The total loss is equal to the weighted sum of $L_{ae}$ and $L_{ls}$, with a weight of $\frac{h \times w \times d}{N_{ls}}$ for $L_{ls}$ and of 1.0 for $L_{ae}$, following the method proposed in (Li et al., 2020).

## 3. Experiments and Results

### 3.1. Clinical data

The data used in this study consists of T1-weighted brain MRI scans of 2118 predominantly healthy adults (1029 males, 1089 females) aged between 21 and 82 years (mean: $51 \pm 14$), acquired on a single scanner using the same scanning parameters. The participants were randomly selected in the region of Pomerania, Germany, as part of the Study of Health In Pomerania (SHIP) (Völzke et al., 2011). Additionally, the IXI database (https://brain-development.org/ixi-dataset/) containing 563 T1-weighted scans of healthy adults aged between 20 and 86 years (mean: $49 \pm 16$), collected on 3 different sites, was used as an independent test set.

### 3.2. Preprocessing

All datasets were first registered to the MNI brain atlas (Mazziotta et al., 2001) with an affine transformation and bias field corrected using ANTs (Tustison et al., 2010; Avants et al., 2011). Three-dimensional patches around the ventricles region, of size $71 \times 93 \times 39$, were cropped in the T1-weighted MRI datasets. This region was chosen because the aging effects are highly visible in the ventricles and it was previously identified as one of the most important region for biological brain age prediction tasks (Levakov et al., 2020). We choose

to use smaller patches instead of the full brain image in this initial study to considerably reduce the computational time, resources, and number of datasets needed to train the model. Finally, the patches were centered and scaled using the mean and the standard deviation of the grey values of each dataset.

### 3.3. Model training

The autoencoder architecture used in this work was adapted from a state-of-the-art age regression CNN (Cole et al., 2017). More specifically, the encoder is made of three consecutive blocks each including: two three-dimensional convolutional layers with $(3{\times}3{\times}3)$ kernels and ReLu activation, one batch normalization layer, and one $(2{\times}2{\times}2)$ max pooling layer. The convolutional layers from the first block have eight filters each, and this number is doubled for each subsequent block. Finally, a convolutional layer with a $(3{\times}3{\times}3)$ kernel, three filters, and ReLu activation reduces the latent space size to 1620 units. The decoder is made of the same architecture as the encoder but is mirrored and ends with one additional convolutional layer with one filter that outputs the reconstructed MRI scan. The latent space disentanglement module is made of two dense layers of 1620 units with shared weights, guaranteeing their invertibility.

The data were randomly split into a training, a validation, and a test set containing respectively 1535, 383, and 200 datasets. In a first step, the autoencoder alone was pretrained using a batch size of 28, and the Adam optimizer with a learning rate of 0.005 for 400 epochs. Then, the latent space representations of the training and validation datasets were saved and used to train the latent space disentanglement module with a batch size of 128 and a learning rate of 0.001, over 3000 epochs. Finally, joint training of the latent space disentanglement module and the autoencoder was performed to fine-tune the weights of both parts using a smaller learning rate of 0.0001.

The generative component of the model uses a Gaussian mixture model with 10 components and full covariance. The Gaussian mixture model representing the distribution of the age-unrelated components was fitted using the training datasets. More specifically, the datasets were passed through the encoder and latent space disentanglement module and their age-unrelated components were used to train the Gaussian mixture model.

Our code is available online at https://github.com/pmouches/Brain-Age-Prediction-and-Age-Conditioned-Template-Generation.

### 3.4. Age prediction

The age prediction ability of the final model was tested on 200 hold out datasets from the SHIP database and 563 datasets from the IXI database. More precisely, the results of the model with and without joint training of the autoencoder and the latent space disentanglement module were compared. Therefore, the age component of the disentangled latent space was extracted and compared to the chronological age of each subject using the mean absolute error and the coefficient of determination ($R^2$). For baseline comparison of the results, a ground truth CNN model with the same architecture as the encoder (Cole et al., 2017) and an additional one-unit dense layer with a linear activation to output the predicted age was trained, validated, and tested on the same datasets (Table 1).

Table 1: Age prediction model accuracy comparison. MAE: Mean absolute error.

| Model | MAE (years) | | $R^2$ | |
|---|---|---|---|---|
| | **SHIP** | **IXI** | **SHIP** | **IXI** |
| INN (Wilms et al., 2020) | 5.05 | 6.95 | NA | NA |
| Ground truth CNN | 5.25 | 8.52 | 0.756 | 0.698 |
| Proposed model without joint training | 5.77 | 8.65 | 0.690 | 0.667 |
| Proposed model with joint training | **4.95** | **6.97** | **0.780** | **0.759** |

Paired t-tests were used to compare the models mean absolute errors. For the tests on the SHIP database, the difference between the ground truth CNN and the proposed model with joint training is not significant (p=0.28). However, the proposed model performs significantly better with joint training than without (p=0.003). For the test on the IXI database, the model with joint training performed significantly better than the ground truth CNN ($p < 10^{-7}$) and than the model without joint training ($p < 10^{-19}$). Additionally, the results from Wilms et al. (2020) who also used the SHIP database to train and test their INN, as well as the data from the IXI database as an independent test set, are reported for state-of-the-art comparison.

### 3.5. Age-specific template generation

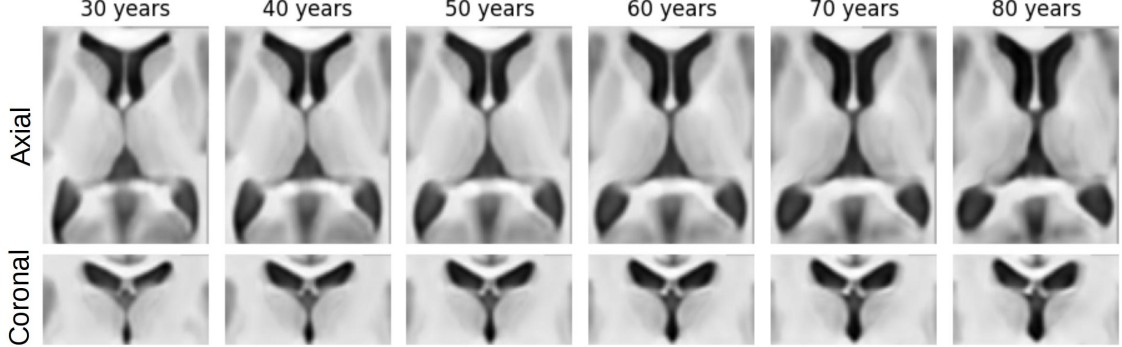

Figure 2: Selected slices of the age-specific templates for 30 to 80 years.

To showcase the ability of our model to disentangle age-related variations from age-unrelated anatomical variations, age-specific templates were generated by averaging the components unrelated to age of the disentangled latent space over all training samples. The age-related component was then fixed to specific ages (30 to 80 years) and the corresponding images were reconstructed (see Figure 2). The templates show realistic increasing volume of the ventricles with aging due to brain atrophy.

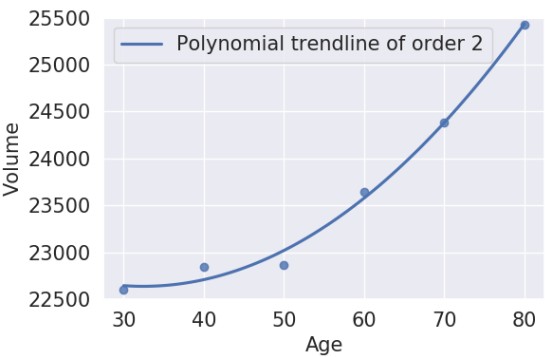

Figure 3: Lateral ventricle volume (in mm³) within each age-specific template.

Additionally, the lateral ventricles in each age-specific template were segmented using a region growing method and manual correction to estimate their volume in order to provide a quantitative assessment of the generated templates (see Figure 3).

### 3.6. Simulated subject-specific aging

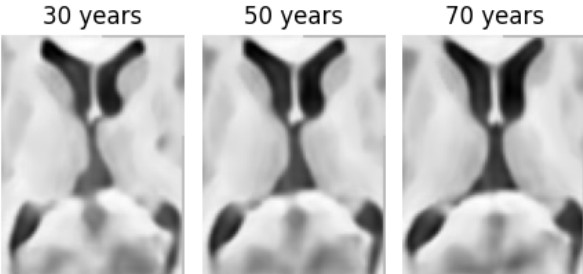

Figure 4: Subject-specific aging example: simulated MRI datasets for a given subject at the age of 30, 50, and 70 years showing natural age-related increased ventricular volume.

In addition to generating age-conditioned templates with an average anatomy, our model is also capable of simulating subject-specific aging effects. More specifically, this is done by sampling age-unrelated components from the Gaussian mixture model, and setting the age-related component to different values. Those latent space representations are then fed through the latent space disentanglement module and the decoder (Figure 1 B) to generate the simulated images. An example can be seen in Figure 4 where the age-related component was set to 30, 50, and 70 years, and the age-unrelated components remained constant, to simulate the aging process for a specific subject. The simulated datasets show realistic aging effects with increasing volume of the ventricle while the components unrelated to age (e.g. shape difference in the left and right inferior part of the ventricles) remain unchanged.

## 4. Discussion and conclusion

In this paper, we propose a novel approach to estimate biological brain age, generate age-specific MRI templates, and simulate MRI scans conditioned on age. Our approach employs well-known deep neural network techniques and results in competitive age prediction and realistic MRI simulation using a single model. We showed that using a deterministic model leads to comparable results described in literature, in which more complex models were used (Zhao et al., 2019; Wilms et al., 2020).

The age prediction estimation of the proposed method is comparable to the results of the baseline CNN model for the SHIP data, and better for the independent test data, highlighting the generalizability of the model. The results of previous models achieving the same task on similar data but using the full brain instead of a cropped patch (Wilms et al., 2020) are also comparable to ours. Additionally, the joint training significantly improves the predictions (Table 1), illustrating the importance of simultaneously training the autoencoder and the latent space disentanglement module, as the deepest layers of the autoencoder can adapt to generate better age disentanglement without deteriorating the image reconstruction quality.

The age-specific templates (Figure 2) show natural age-related shape variations with an increased ventricular volume and wider sulci associated with increasing age, demonstrating the age disentanglement ability of the model. The model was also able to capture the non-linear aging trend as shown by the non-linear volume increase of the lateral ventricles in Figure 3, in line with previously observed trends (Pfefferbaum et al., 2013).

Finally, the simulated MRI scans (Figure 4) look realistic and the age-related changes are visible while the subject-specific anatomy remains nearly unchanged. This indicates a successful disentanglement of the age-related and unrelated factors of variation.

In general, our results show that the proposed model is able to generate realistic images while also producing highly accurate age predictions. In future work, we plan to increase our training and testing database, as well as to improve the model training parameters in order to achieve an even more accurate latent space disentanglement. Additionally, we plan to investigate counterfactuals, which can easily be generated with the current model architecture and bring explainability to the biological brain age prediction task. Potential further applications of the model include disentanglement of other factors than age, or of several factors simultaneously.

## 5. Acknowledgments

This work was supported by the Canadian Open Neuroscience Platform, the Canada Research Chairs Program, the River Fund at Calgary Foundation and the Hotchkiss Brain Institute.

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
