# OpenReview forum: "Unifying Brain Age Prediction and Age-Conditioned Template Generation with a Deterministic Autoencoder"
_MIDL.io/2021/Conference — MIDL 2021_

### Official Review · AnonReviewer3 · 2021-02-25

**Confidence:** 4
**Preliminary Rating:** 2
**Recommendation:** Poster
**Final Rating:** 2

**Summary:**

The paper at hand addresses the problem of simultaneous brain age estimation and age-conditioned image generation. The authors propose the use of a normal deterministic autoencoder, coupled with a subspace projection method to disentangle age and other components in the latent space. Generating MRI scans is achieved by fixing the age component and randomly sampling from the other components that are modeled as a GMM. Evaluation is performed on a dataset of 2118 MRI scans of healthy patients. In terms of results, the authors show age-specific templates by averaging the latent space of non-age components across the training set and generating images from that. Also, the authors show age regression performance and one example of subject-specific scan generation for different ages.

**Strengths:**

•	The paper is well-written and easy to follow

•	The motivation for the performed work is clearly conveyed to the reader

•	Related work and prior art is discussed and put into the context of this work

•	The results are moderately interesting


**Weaknesses:**

•	From a methodology point of view, novelty is limited. The authors use the idea of Gosh et al. to use deterministic auto-encoders instead of VAEs. This is combined with an idea for latent space separation (in this case, applied to age) by Li et al. Thus, I would put this paper towards application/method validation, as it shows the application of existing methods to the problem of brain age estimation and simulated aged scan generation.

•	However, if this paper is treated as an application paper, the contributions are somewhat limited. I would have expected a more extensive evaluation. In particular, the only comparison to prior work is presented in Table 1 for age regression with a comparison to a single paper (same research group as this paper).

•	Furthermore, generating realistic, aged scans is one of the primary contributions. However, there is no qualitative comparison to other generative models, in particular, VAEs. Also, there is no comparison to results from Wilms et al. which was used as a comparison for age regression in Table1 (which is from the same research group).

•	The authors cite Zhao as a method that solves the same problem (brain age estimation + generating simulated aged scans) with VAEs. However, there is no comparison to their method.

•	The value of the experiments in Section 3.5 is not entirely clear to me. The main takeaway seems to be: the model learned to represent the training set correctly.


**Deanonymize Review:**

no

**Detailed Comments:**

See weaknesses. From my point of view, more comparisons to other methods would be needed. In particular, the comparison to VAEs would be crucial, as this is an existing method that was already applied to the problem that the authors aim to solve.

**Final Rating Justification:**

The authors provide answers to the concerns I raised. However, I am not fully convinced that comparison to other methods is not required. This is emphasized by comments from AnonReviewer4 who raises valid points about more advanced papers that address (in parts) the same problem.

**Justification Of The Preliminary Rating:**

As indicated above, I see this paper as the application of existing methods to a different problem. Therefore, I would expect experiments that compare this method to other methods that have been used to solve this problem (in particular VAEs). Since this is not the case, I recommend rejecting the paper in its current state.

**Paper Type:**

validation/application paper

**Questions To Address In The Rebuttal:**

I think adding more experiments is out of scope for the rebuttal. The authors could however try to convince me, that a comparison to VAEs is not required. Furthermore, the authors could clarify the value of the experiments in Section 3.5.

**Special Issue:**

no

---

> ### Author Response · Authors · 2021-03-17
> **Part 1 - Point-by-point response to Anon Reviewer3**
>
> We thank the reviewer for his comments and thoughtful suggestions. We provide below a point-by-point response.
>
> **From a methodology point of view, novelty is limited. The authors use the idea of Gosh et al. to use deterministic auto-encoders instead of VAEs. This is combined with an idea for latent space separation (in this case, applied to age) by Li et al. Thus, I would put this paper towards application/method validation, as it shows the application of existing methods to the problem of brain age estimation and simulated aged scan generation.**
>
> In general, we agree that this paper proposes to combine ideas from the machine learning literature to solve a challenging problem in neuroimaging. However, we consider this combination as a novel contribution that has never been proposed before (neither in machine learning nor medical imaging) and also strongly believe that applying this concept to the brain aging problem qualifies as a novel contribution. More precisely, Gosh et al. [1] and Li et al. [2] only conduct experiments on fairly simple, standard databases (Gosh et al. : MNIST, CIFAR10, CELEBA; Li et al.: CELEBA). In contrast, we show in our work that combining and adapting both methods allows us to (1) handle fairly complex 3D medical data and (2) successfully disentangle age-related and age-unrelated information in brain MR images.
>
> **However, if this paper is treated as an application paper, the contributions are somewhat limited. I would have expected a more extensive evaluation. In particular, the only comparison to prior work is presented in Table 1 for age regression with a comparison to a single paper (same research group as this paper).**
>
> First of all, we would like to clarify that we do not only provide results for our previously published method (Wilms et al. [5]) but also report results for a state-of-the-art, standard CNN-based age-prediction approach (Cole et al.; named ‘Ground truth CNN’ in Tab. 1). In addition, we followed Reviewer 1’s suggestion and also added additional results using an independent test set (IXI data) to Tab. 1 (see updated paper).
> A more extensive comparison with the literature is not straightforward to do as each study uses different databases, with subjects from different age ranges, which result in mean absolute errors (MAE) comparing the predicted to the true age ranging from about 2 to 6 years. The paper that is the most comparable to our results [3] also used a CNN and data from the SHIP study to test it. They reported a MAE of 4.12 [3], which shows that the SHIP database is challenging compared to other databases such as the UK Biobank with which several papers reported MAE of ~2-3 years [4]. Moreover, in [3], ~11,000 datasets were used to train the model and the full brain image while we are only using a small patch here. Thus, our results for the age prediction task seem reasonable.
>
> **Furthermore, generating realistic, aged scans is one of the primary contributions. However, there is no qualitative comparison to other generative models, in particular, VAEs. Also, there is no comparison to results from Wilms et al. which was used as a comparison for age regression in Table1 (which is from the same research group).**
>
> We agree that our paper could benefit from a better qualitative or quantitative comparison of the generated images. However, approaches achieving the same task are rather sparse in the literature so far. Our generated images would not be qualitatively comparable to the ones from Wilms et al. [5], as the model proposed in [5] generates deformation fields that are then applied to a brain atlas, and not grey value images as done here. Thus, they do not work with the voxel intensity values directly but with a simpler representation. On the other hand, the only VAE approach [6] that has been proposed so far for unified brain age prediction/image generation cannot be used for subject-specific aging. Nevertheless, in order to enable a better (quantitative) assessment of the generated images and the age disentanglement ability of the model, we provide the age-dependent ventricular volume information (see Fig. 3), which shows a realistic increasing trend.

---

> ### Author Response · Authors · 2021-03-17
> **Part 2 - Point-by-point response to Anon Reviewer3**
>
> **Part 2 - Please see Part 1 first**
>
> **The authors cite Zhao as a method that solves the same problem (brain age estimation + generating simulated aged scans) with VAEs. However, there is no comparison to their method.**
>
> We agree with the reviewer that the approach from Zhao et al. [6] is somewhat similar to ours and would be a good comparison. Unfortunately, we tried to run their open source code but have so far been unsuccessful doing so with our data. However, by visually comparing their templates (see [6], Fig. 2) to ours (Fig. 2), we can notice that ours provide better image quality and sharpness (they also used downsampled images). Additionally, our model allows us to simulate patient specific aging, which is impossible with the VAE as proposed in [6].
>
> **The value of the experiments in Section 3.5 is not entirely clear to me. The main takeaway seems to be: the model learned to represent the training set correctly.**
>
> Experiment 3.5 has two components: (1) It shows that the model is able to disentangle the age from the age-unrelated components. Each template is generated using the age-unrelated components from all MRI scans, and only the age component is changed. Thus, it shows that changing only the age value leads to images that faithfully represent the (average) aging trend in our population. (2) This experiment also illustrates that our model is able to solve a problem relevant in, for example, neuroscience research where age-matched average brain templates are frequently used for spatial normalization of image datasets prior to voxel-based analyses (see for example [7]). In comparison to more traditional registration-based template generation approaches, our model is a continuous one and able to generate a brain template for any specific age value even those not available during training (interpolation and extrapolation capabilities).
>
> *References:*
>
> [1] Ghosh, Partha, et al. "From variational to deterministic autoencoders." arXiv preprint arXiv:1903.12436 (2019).
>
> [2] Li, Xiao, et al. "Latent space factorisation and manipulation via matrix subspace projection." International Conference on Machine Learning. PMLR, 2020.
>
> [3] Bashyam, Vishnu M., et al. "MRI signatures of brain age and disease over the lifespan based on a deep brain network and 14 468 individuals worldwide." Brain 143.7 (2020): 2312-2324.
>
> [4] Peng, Han, et al. "Accurate brain age prediction with lightweight deep neural networks." Medical Image Analysis 68 (2021): 101871.
>
> [5] Wilms, Matthias, et al. "Bidirectional Modeling and Analysis of Brain Aging with Normalizing Flows." Machine Learning in Clinical Neuroimaging and Radiogenomics in Neuro-oncology. Springer, Cham, 2020. 23-33.
>
> [6] Zhao, Qingyu, et al. "Variational autoencoder for regression: Application to brain aging analysis." International Conference on Medical Image Computing and Computer-Assisted Intervention. Springer, Cham, 2019.
> [7] Rajashekar, Deepthi, et al. "High-resolution T2-FLAIR and non-contrast CT brain atlas of the elderly." Scientific Data 7.1 (2020): 1-7.

---

### Official Review · AnonReviewer1 · 2021-03-05

**Confidence:** 4
**Preliminary Rating:** 3
**Recommendation:** Poster

**Summary:**

Brain age estimation from MRI scans using autoencoders and latent space disentanglement learning is proposed. The method is evaluated on 2118 healthy adults. The paper is well written and easy to read. Evaluation is performed in a relatively large dataset. the authors show that, by using their proposed method, they are able to generate realistic MRI images while also producing highly accurate age predictions. With some improvements, the paper can be accepted as it will create healthy discussions

**Strengths:**

The paper is well written and easy to read.
Evaluation is performed on a relatively large dataset.
Comparison against prior work is performed.
Results show that the method can generate realistic images with age information.


**Weaknesses:**

More details should be included on how the training and testing data was split (see detailed comments section).
Prior work evaluation on the authors' dataset is not reported.
Autoencoder is based on prior work.


**Deanonymize Review:**

no

**Detailed Comments:**

How does the proposed method compare against the below work where age estimation errors of 3.58 are reported vs the 4.95 reported in this work.

Jónsson BA, Bjornsdottir G, Thorgeirsson TE, Ellingsen LM, Walters GB, Gudbjartsson DF, Stefansson H, Stefansson K, Ulfarsson MO. Brain age prediction using deep learning uncovers associated sequence variants. Nature communications. 2019 Nov 27;10(1):1-0.

What do authors mean by raw MRI scan? Are they talking about the k-space information before reconstruction of the MRI scan? If this is not the case, if they are using MRI data as shown in Fig1, please delete the word ‘raw’ from the text as it creates confusion.

Why is the cropping of the MRI data required? Is this related to getting rid of some image information not necessary for age estimation (reducing the feature space) or is the size of the ventricles related to age? More information should be provided.

MAJOR: The authors should provide more information on how the dataset split was performed and if the process was repeated (k-fold evaluation).

Have the authors re-evaluated the method proposed by Wilms et all (INN) on their dataset or are they reporting what is reported on that paper? I am assuming it is the later one as the number is the same. This is an unfair comparison as the two datasets are different and the INN method has a larger testing dataset.

As in Wilms et al 2020 can the authors provide evaluation results on publicly available data  https://brain-development.org/ixi-dataset/ (use this as test data only).
This will provide a chance to compare the proposed method against that work using the same dataset which would increase the strength of this work.


**Justification Of The Preliminary Rating:**

The paper is well written and easy to read. I would like to see the authors re-evaluate their method using the publicly available dataset  (https://brain-development.org/ixi-dataset/) as this would provide a direct comparison with the recent work of Wilms et all (INN) and improve the strength of the work (if the results are significantly improved over Wilms et al work)

**Paper Type:**

both

**Questions To Address In The Rebuttal:**

use the publicly available data (https://brain-development.org/ixi-dataset/) to re-evaluate their method.
The authors should provide more information on how the dataset split was performed and if the process was repeated (k-fold evaluation).



**Special Issue:**

no

---

> ### Author Response · Authors · 2021-03-17
> **Part 1 - Point-by-point response to AnonReviewer1**
>
> We thank the reviewer for his comments and thoughtful suggestions. We provide below a point-by-point response.
>
> **How does the proposed method compare against the below work where age estimation errors of 3.58 are reported vs the 4.95 reported in this work.**
>
> We thank the reviewer for mentioning the work from Jonsson et al. Recently, a lot of different approaches have been proposed to predict the brain age. However, it is difficult to directly compare the obtained errors as there is no established benchmark dataset publicly available. Hence, each study uses a different number of datasets, with a different age range, and different information. For example, Jonsson et al. use the sex and scanner information in their CNN, while we don’t. The most comparable study we found so far using the same database (SHIP) to test their model is [1]. They also use a CNN and report a mean absolute error (MAE) of 4.12 [1], which confirms that the SHIP database is especially challenging. Moreover, in [1], ~11,000 datasets were used to train the model, as well as the full brain image while we are only using a small patch here. We, therefore, argue that our results for the age prediction task are within the range of recently described results.
>
> **What do authors mean by raw MRI scan? Are they talking about the k-space information before reconstruction of the MRI scan? If this is not the case, if they are using MRI data as shown in Fig1, please delete the word ‘raw’ from the text as it creates confusion.**
>
> Raw MRI scans referred to the MRI data as shown in figure 1A. Thus, by raw, we meant that there was minimal preprocessing steps done on the scans, compared to some previous studies who extracted features from the scans. We would like to thank the reviewer for highlighting this potential source of confusion to us. To solve this issue, we removed the term ‘raw’ from the paper and replaced it by the term ‘minimally pre-processed’.
>
> **Why is the cropping of the MRI data required? Is this related to getting rid of some image information not necessary for age estimation (reducing the feature space) or is the size of the ventricles related to age? More information should be provided.**
>
> The cropping of the images was used in this initial study to reduce the problem complexity while evaluating the feasibility of the proposed approach. Indeed, using a full image requires more computational memory to train a bigger and deeper network, and may require more datasets to optimize the model parameters. Using cropped images is a method that has also been used in past work for similar tasks [2, 3]. Within this context, it should be highlighted that the ventricle region is the most informative region for brain age estimation as even small atrophy changes throughout the brain lead to visible expansion of the ventricles to compensate for the lost brain tissue.
>
> *References:*
>
> [1] Bashyam, Vishnu M., et al. "MRI signatures of brain age and disease over the lifespan based on a deep brain network and 14 468 individuals worldwide." Brain 143.7 (2020): 2312-2324.
>
> [2] Zhao, Qingyu, et al. "Variational autoencoder for regression: Application to brain aging analysis." International Conference on Medical Image Computing and Computer-Assisted Intervention. Springer, Cham, 2019.
>
> [3] Bintsi, Kyriaki-Margarita, et al. "Patch-Based Brain Age Estimation from MR Images." Machine Learning in Clinical Neuroimaging and Radiogenomics in Neuro-oncology. Springer, Cham, 2020. 98-107.

---

> ### Author Response · Authors · 2021-03-17
> **Part 2 - Point-by-point response to AnonReviewer1**
>
> **Part 2 - Please see Part 1 first**
>
> **MAJOR: The authors should provide more information on how the dataset split was performed and if the process was repeated (k-fold evaluation).**
>
> The data split was performed randomly, using 1535, 383 and 200 datasets for training, validation, and testing, respectively. No cross-validation was performed. The age prediction accuracy results presented in section 3.4 were computed on the 200 hold-out testing datasets. We have also added the results when applied to a completely independent test set (IXI) database (see comment and corresponding answer below), which confirms the generalizability of the proposed method.
>
> **Have the authors re-evaluated the method proposed by Wilms et al. (INN) on their dataset or are they reporting what is reported on that paper? I am assuming it is the later one as the number is the same. This is an unfair comparison as the two datasets are different and the INN method has a larger testing dataset.**
>
> The reported comparison results are taken from the paper by Wilms et al directly where a slightly different test/training/validation split was used. Although we agree that this makes them not directly comparable to our results, we believe that listing the INN results still helps to interpret our new results as the INN model aims to solve the same bidirectional task as the novel model proposed here. We, furthermore, think that it is quite interesting to see that we are able to achieve results in the same range as Wilms et al. by using less training data, and cropped images. We believe that this illustrates the potential of the proposed approach.
>
> **As in Wilms et al 2020 can the authors provide evaluation results on publicly available data https://brain-development.org/ixi-dataset/ (use this as test data only). This will provide a chance to compare the proposed method against that work using the same dataset which would increase the strength of this work.**
>
> We thank the reviewer for this suggestion. We downloaded and pre-processed the data from IXI, as done for the SHIP data. The reported mean absolute errors on the IXI data are the following: Ground truth CNN: 8.52; Proposed model without joint training: 8.65; Proposed model with joint training: 6.97; with the proposed model with joint training performing significantly better than the two others. This result is comparable to the results of the INN as described in Wilms et al. on the same IXI data (6.95). These results have been added to the paper (see updated Table 1). One potential reason why the error is higher than on the SHIP data is related to the scanning parameters: The data from SHIP were acquired on a single scanner with the same scanning parameters, while IXI contains data from three sites, with scanning parameters different from the SHIP data. The parameters affect the image intensities and our preprocessing steps might not be sufficient to fully account for these differences. In future work, we will train our model with data from different sites to make it more robust.

---

### Official Review · AnonReviewer4 · 2021-03-07

**Confidence:** 5
**Preliminary Rating:** 1

**Summary:**

This paper seeks to improve brain age prediction by explicitly disentangling brain age factors from confound variables in the latent space of an auto encoder. They make predictions only based on a small patch of data surrounding the ventricles and compare against a conventional CNN to show improvements.


**Strengths:**

The authors correctly assert that accurate brain age prediction is confounded by natural heterogeneity of brain shape and other factors and propose a sound approach to disentangling the latent space of a deep model. Their approach clearly improves brain age prediction on their dataset and relative to a straightforward CNNs. The paper is well motivated and clearly presented


**Weaknesses:**

While this paper is on the correct track there were two more advanced approaches solving related problems which were published in NeurIPS this year

https://papers.nips.cc/paper/2020/hash/0987b8b338d6c90bbedd8631bc499221-Abstract.html
https://papers.nips.cc/paper/2020/hash/56f9f88906aebf4ad985aaec7fa01313-Abstract.html

These go further to detect subject specific features of ageing across all the brain or generate causal models of ageing in ways that would generalise to and beyond brain age prediction. So this paper would need to position itself relative to these papers.
The main limitations of this paper therefore is that it is not a generative or causal model so it cannot generate novel examples or counterfactual explanations, nor generate sharp reconstructions like a GAN might e.g https://slideslive.com/38942415/using-stylegans-for-visual-interpretability-of-deep-learning-models-on-medical-images

In terms of results, the model is only trained on patches around the ventricles, which is quite a limitation of the model given that previous approaches can be trained on whole brains. It also means it’s impossible to tell whether it can disentangle heterogeneous effects around the cortex but one suspects not.

Finally, conditional template generation has also been previously considered in Dalca, Adrian V., et al. "Learning conditional deformable templates with convolutional networks." arXiv preprint arXiv:1908.02738 (2019). (published Neurips 2019)




**Deanonymize Review:**

no

**Detailed Comments:**

The notation in section 2.1 seems more than an abuse of notation, it is actually quite confusing for what is a simple concept. Also I think d(z_i) should equal \hat{X}_i?


**Justification Of The Preliminary Rating:**

Unfortunately, despite being well presented with promising experiments, the methods in this paper are not presenting novel ideas and it seems unlikely the method would perform well against the state of the art.

**Paper Type:**

methodological development

**Questions To Address In The Rebuttal:**

Please frame this work in the context of recent peer reviewed literature in the field

**Special Issue:**

no

---

> ### Author Response · Authors · 2021-03-17
> **Part 1 - Point-by-point response to AnonReviewer4**
>
> We thank the reviewer for appreciating our disentangling approach for brain age prediction and the many insightful comments. However, we strongly disagree with the overall assessment of our paper regarding its novelty and would like to use this opportunity to clarify some aspects that this work is indeed novel.
>
> **While this paper is on the correct track there were two more advanced approaches solving related problems which were published in NeurIPS this year**
> **https://papers.nips.cc/paper/2020/hash/0987b8b338d6c90bbedd8631bc499221-Abstract.html**
> **https://papers.nips.cc/paper/2020/hash/56f9f88906aebf4ad985aaec7fa01313-Abstract.html**
> **These go further to detect subject specific features of ageing across all the brain or generate causal models of ageing in ways that would generalise to and beyond brain age prediction. So this paper would need to position itself relative to these papers. The main limitations of this paper therefore is that it is not a generative or causal model so it cannot generate novel examples or counterfactual explanations, nor generate sharp reconstructions like a GAN might e.g https://slideslive.com/38942415/using-stylegans-for-visual-interpretability-of-deep-learning-models-on-medical-images**
>
> We would like to thank the reviewer to point out those NeurIPS papers. While they also build (generative) models related to brain aging and related mechanisms, none of them actually solves the problems we are tackling here in a unified way in 3D: (1) brain age prediction, (2) age-conditioned template generation, and (3) subject-specific aging synthesis.
> Before discussing the specific differences between our work and the papers mentioned above, we would like to clarify that our model is (1) a generative model while (2) also being capable of generating meaningful counterfactuals.
>
> *Generative model:* We agree that a standard, deterministic autoencoder is not a generative model as a mechanism to sample new data/subjects. This is the reason why we introduce a Gaussian Mixture Model (GMM) that models the complex distribution of age-unrelated anatomical components in the disentangled latent space (see last paragraph of Sec. 3.3 & Fig. 1). This GMM essentially learns/models the age-unrelated inter-subject variability in our training dataset. Sampling from this GMM while fixing the age component and feeding the new data through the decoder part of our network generates images of fake subjects with a specified brain age (see Sec 3.6; Fig. 4). Changing the age component, for example, also allows us to generate subject data for age values not available during training (interpolation or extrapolation beyond the training data). We, therefore, believe that our model qualifies as a generative modeling approach.
>
> *Counterfactuals:* While we do not use a traditional model of causality like a structural causal model (SCM), our approach is still able to generate counterfactuals for the brain aging problem. In our age prediction scenario, counterfactual images are, for example, images of the same subject that would lead to a different age prediction than the original, real image (e.g., older or younger versions of this subject). Such counterfactual images can be generated by first fixing the age-unrelated anatomical components in the disentangled latent space derived from the original images. Then, the age variable is manipulated (younger or older) and the data is fed through the decoder part of our network to generate a corresponding image. We explicitly exploit this mechanism in our paper in Sec. 3.6 to simulate subject-specific aging based on sampled age-unrelated anatomical components (the same age-unrelated anatomical components were used to reconstruct 3 images with different ages). The same concept could also easily be used to explain the decisions made by our model.
>
> *Generalization beyond brain aging:* While we only apply our model to the brain aging process, the general idea could also be easily applied to other problems in neuroimaging and beyond. A simple and straightforward extension would be to consider additional variables/factors during disentanglement (e.g., sex or disease status) to be able to predict and independently manipulate them.

---

> ### Author Response · Authors · 2021-03-17
> **Part 2 - Point-by-point response to AnonReviewer4**
>
> **Part 2 - Please see Part 1 first**
>
> *Differences to Pawlowski et al. @ NeurIPS2020:* We agree with the reviewer that this paper goes further than what we do in terms of disentanglement of different factors as they are explicitly modeling causal relationships using a causal model. This is definitely a really powerful approach as it covers all aspects of Pearl’s ladder of causation. While they also use brain modeling from MRI data as one of their scenarios, they do not use their model for actual age prediction. Furthermore, as far as we understand the paper, their approach seems to be computationally quite expensive, which limits their analysis to isolated 2D slices. This computational factor, in particular, is one of the advantages of our work as it is easy and inexpensive to train. However, we have to admit that we also had to restrict our analysis to cropped 3D patches due to lack of GPU memory on our training hardware.
>
> *Differences to Bass et al. @ NeurIPS2020:* While we agree with the reviewer that this is also an amazing paper, we do not agree with the assessment that this work goes further than ours. The authors of that paper also use a more traditional approach for disentangling factors (no SCM as in Pawlowski et al.) and they seem to focus mainly on the interpretability of the learned classifier. In contrast, our approach is not restricted to classification problems as we are solving a challenging, continuous regression problem here and while we do not explicitly exploit our model’s capability to explain its decisions, generating heatmaps based on different counterfactuals would allow us to also explain our regressor in a similar way as they do.
>
> *Differences to Schutte et al. @ NeurIPSWS2020:* We agree that our approach is most likely not capable of generating images that are as sharp as those generated by GANs. However, we argue that (1) our images (templates and counterfactuals) are quite sharp and (2) that their quality is, for example, sufficient to interpret the regressor in a more meaningful way than a GradCAM heatmap would allow. Furthermore, the advantage of our approach in contrast to a GAN as used in Schutte et al. is that our model is a unified, consistent one with both directions readily available and jointly trained.
>
> **In terms of results, the model is only trained on patches around the ventricles, which is quite a limitation of the model given that previous approaches can be trained on whole brains. It also means it’s impossible to tell whether it can disentangle heterogeneous effects around the cortex but one suspects not.**
>
> We agree and acknowledge that using patches around the ventricles in this work is a limitation when the goal is to translate such a model into clinical practice. However, we are confident that with additional optimizations of our architecture, we will be able to use full 3D image data. In addition, we disagree with the assessment that (all) previous generative approaches for brain aging can be trained on whole brains. Many deep learning-based papers initially restrict their analysis to 2D slices or 3D patches to handle the computational challenges involved with building large population-based models. This is, for example, true for the work by Zhao et al. (cited in our paper; 3D patches) and the work from Pawlowski et al. (see above, 2D slices).
>
> **Finally, conditional template generation has also been previously considered in Dalca, Adrian V., et al. "Learning conditional deformable templates with convolutional networks." arXiv preprint arXiv:1908.02738 (2019). (published Neurips 2019)**
>
> Dalca et al. also build a deep learning-based model of brain aging. However, their approach is neither able to perform age prediction, nor do they provide a mechanism to sample new data (e.g., counterfactuals) aside from the template generation process.
>
> **The notation in section 2.1 seems more than an abuse of notation, it is actually quite confusing for what is a simple concept. Also I think d(z_i) should equal \hat{X}_i?**
>
> We apologize for any confusion that our notation has caused and we would be happy to modify it based on additional feedback. Thank you for spotting the inconsistency with respect to \hat{X}_i. Indeed, it should be d(z_i)=\hat{X}_i \approx X_i, which has beeen corrected in paper.

---

### Meta-Review · Area_Chair1 · 2021-03-28

**Recommendation:** Accept (Poster)

**Metareview:**

While reviewers pointed out a desire for quantitative comparison to additional other work (some comparison already does exist in the paper), this paper proposes a new approach that allows for brain age prediction, age-conditioned template generation, and subject-specific aging simulation and demonstrates results on a robustly sized dataset. Based on helpful suggestions from the preliminary review, they also added a secondary dataset during rebuttal, which further strengthened the experimental results section. As such, I believe this is a meaningful study that would be of interest to the MIDL community and contribute to the brain age prediction literature.

**Paper Type:**

both

---

### Decision · Program_Chairs · 2021-03-31

Accept